# Risk assessment models for venous thromboembolism in hospitalised adult patients: a systematic review

Abdullah Pandor ![ORCID],[1] Michael Tonkins,[1] Steve Goodacre ![ORCID],[1] Katie Sworn,[1] Mark Clowes,[1] Xavier L Griffin ![ORCID],[2] Mark Holland,[3] Beverley J Hunt,[4] Kerstin de Wit ![ORCID],[5] Daniel Horner ![ORCID] [6]

[1]ScHARR, The University of Sheffield, Sheffield, UK
[2]Barts and The London School of Medicine and Dentistry, Queen Mary University of London, London, UK
[3]Department of Clinical and Biomedical Sciences, University of Bolton, Bolton, UK
[4]Department of Haematology, Guy's and St Thomas' NHS Foundation Trust, London, UK
[5]Department of Medicine, McMaster University, Hamilton, Ontario, Canada
[6]Emergency Department, Salford Royal NHS Foundation Trust, Salford, UK

**Correspondence to**
Abdullah Pandor;
a.pandor@sheffield.ac.uk

## ABSTRACT

**Introduction** Hospital-acquired thrombosis accounts for a large proportion of all venous thromboembolism (VTE), with significant morbidity and mortality. This subset of VTE can be reduced through accurate risk assessment and tailored pharmacological thromboprophylaxis. This systematic review aimed to determine the comparative accuracy of risk assessment models (RAMs) for predicting VTE in patients admitted to hospital.

**Methods** A systematic search was performed across five electronic databases (including MEDLINE, EMBASE and the Cochrane Library) from inception to February 2021. All primary validation studies were eligible if they examined the accuracy of a multivariable RAM (or scoring system) for predicting the risk of developing VTE in hospitalised inpatients. Two or more reviewers independently undertook study selection, data extraction and risk of bias assessments using the PROBAST (Prediction model Risk Of Bias ASsessment Tool) tool. We used narrative synthesis to summarise the findings.

**Results** Among 6355 records, we included 51 studies, comprising 24 unique validated RAMs. The majority of studies included hospital inpatients who required medical care (21 studies), were undergoing surgery (15 studies) or receiving care for trauma (4 studies). The most widely evaluated RAMs were the Caprini RAM (22 studies), Padua prediction score (16 studies), IMPROVE models (8 studies), the Geneva risk score (4 studies) and the Kucher score (4 studies). C-statistics varied markedly between studies and between models, with no one RAM performing obviously better than other models. Across all models, C-statistics were often weak (<0.7), sometimes good (0.7–0.8) and a few were excellent (>0.8). Similarly, estimates for sensitivity and specificity were highly variable. Sensitivity estimates ranged from 12.0% to 100% and specificity estimates ranged from 7.2% to 100%.

**Conclusion** Available data suggest that RAMs have generally weak predictive accuracy for VTE. There is insufficient evidence and too much heterogeneity to recommend the use of any particular RAM.

**PROSPERO registration number** Steve Goodacre, Abdullah Pandor, Katie Sworn, Daniel Horner, Mark Clowes. A systematic review of venous thromboembolism RAMs for hospital inpatients. PROSPERO 2020 CRD42020165778. Available from https://www.crd.york.ac.uk/prospero/display_record.php?RecordID=165778https://www.crd.york.ac.uk/prospero/display_record.php?RecordID=165778

## Strengths and limitations of this study

► This systematic review provides an up-to-date comprehensive review of risk assessment models for predicting venous thromboembolism in patients admitted to hospital.
► The newly developed PROBAST (Prediction model Risk Of Bias ASsessment Tool) tool was used to evaluate the risk of bias and applicability of the available evidence.
► Heterogeneity in the included studies (participants, inclusion criteria, clinical condition, outcome definition and measurement) and variable reporting of items precluded meta-analysis.
► Limitations of the existing evidence and areas of future research are highlighted.

## INTRODUCTION

Venous thromboembolism (VTE) is an important and life-threatening complication of hospitalisation and illness, and is associated with significant morbidity and mortality.[1] [2] Globally, an estimated 10 million VTE episodes are diagnosed each year; over half of these episodes are associated with hospital inpatients stays and result in significant loss of disability-adjusted life years.[3] [4] Consequently, there has been a substantial and sustained focus on VTE prevention over the last three decades, with good evidence indicating a reduction in morbidity with primary thromboprophylaxis in hospitalised patients.[5–8] Despite this evidence, thromboprophylaxis remains either underused or inappropriately applied.[9]

Risk assessment models (RAMs) have been developed to help stratify the risk of VTE among hospitalised patients.[10] These models use clinical information from the patient's history and examination to identify those with an increased risk of developing VTE who are most likely to benefit from pharmacological prophylaxis. Inappropriate use of VTE

prophylaxis may not reduce VTE rates and may cause unnecessary harm.[11] While RAMs could improve the ratio of benefit to risk and benefit to cost, it is unclear which VTE RAM should be applied to guide decision-making for prophylaxis in clinical practice and thereby optimise patient care.

The current review extends and updates three broadly overlapping existing reviews.[10 12 13] While these reviews identified the use of various (derived and validated) RAMs for VTE in hospitalised patients, they did not find any evidence to suggest which RAM was superior. The aim of this systematic review was to identify primary validation studies (as derivation studies may give an overoptimistic assessment of model performance measures) and determine the accuracy of individual RAMs for predicting the risk of developing VTE in hospital inpatients.

## METHODS

A systematic review was undertaken in accordance with the general principles recommended in the Preferred Reporting Items for Systematic Reviews and Meta-Analyses (PRISMA) statement.[14] This review was part of a larger project on VTE RAMs for hospital inpatients[15] and was registered on the International Prospective Register of Systematic Reviews (PROSPERO) database (CRD42020165778).

### Eligibility criteria

We sought studies evaluating RAMs which could be applied to a general inpatient population (medical, surgical or trauma) rather than disease-specific models. All primary validation studies that evaluated the accuracy (eg, sensitivity, specificity, C-statistic) of a multivariable RAM (or scoring system) for predicting the risk of developing VTE were eligible for inclusion. We selected studies that included validation of the model in a group of patients that were not involved in model derivation. This involved either splitting the study cohort (internal) or using a new cohort (external). The study could have reported derivation of the model but we only used the validation data to estimate accuracy. The study population consisted of hospital inpatients including those who required medical care, undergoing any surgery (excluding day surgery) or received care following an injury. Studies that primarily focused on children (aged under 16 years), women admitted to hospital for pregnancy-related reasons and any patient admitted to a level 2 or above critical care environment (eg, patients requiring more detailed observation or intervention including support for a single failing organ system or postoperative care and those 'stepping down' from higher levels of care) were excluded. These patient groups have VTE risk profiles that differ markedly from the general inpatient population, making the use of a generic model inappropriate.

### Data sources and searches

Potentially relevant studies were identified through searches of five electronic databases including MEDLINE (with MEDLINE In-process and Epub Ahead of Print), EMBASE and the Cochrane Library. The search strategy used free text and thesaurus terms and combined synonyms relating to the condition (eg, VTE in medical inpatients) with risk prediction modelling terms. No language restrictions were used. However, as the current review updated three previous systematic reviews,[10 12 13] searches were limited by date from 2017 (last search date from earlier reviews)[10] to February 2021. Searches were supplemented by hand-searching the reference lists of all relevant studies (including existing systematic reviews); forward citation searching of included studies; contacting key experts in the field; and undertaking targeted searches of the World Wide Web using the Google search engine. Further details on the search strategy can be found in online supplemental appendix S1.

### Study selection

All titles were examined for inclusion by one reviewer (KS) and any citations that clearly did not meet the inclusion criteria (eg, non-human, unrelated to VTE inpatients) were excluded. All abstracts and full-text articles were then examined independently by two reviewers (KS and AP). Any disagreements in the selection process were resolved through discussion or if necessary, arbitration by a third reviewer (SG) and included by consensus.

### Data extraction and quality assessment

Data relating to study design, methodological quality and outcomes were extracted by one reviewer (KS) into a standardised data extraction form and independently checked for accuracy by a second (AP or MT). Any discrepancies were resolved through discussion to achieve agreement. Where differences were unresolved, a third reviewer's opinion was sought (SG). Where multiple publications of the same study were identified, data were extracted and reported as a single study.

The methodological quality of each included study was assessed using PROBAST (Prediction model Risk Of Bias ASsessment Tool).[16 17] This instrument evaluates four key domains: patient selection, predictors, outcome and analysis. Each domain is assessed in terms of risk of bias and the concern regarding applicability to the review (first three domains only). To guide the overall domain-level judgement about whether a study is at high, low or an unclear (in the event of insufficient data in the publication to answer the corresponding question) risk of bias, subdomains within each domain include a number of signalling questions to help judge with bias and applicability concerns. An overall risk of bias for each individual study was defined as low risk when all domains were judged as low; and high risk of bias when one or more domains were considered as high. Studies were assigned an unclear risk of bias if one or more domains were unclear and all other domains were low.

## Data synthesis and analysis

We were unable to perform meta-analysis due to significant levels of heterogeneity between studies (participants, inclusion criteria, clinical condition) and variable reporting of items. As a result, a prespecified narrative synthesis approach[18][19] was undertaken, with data being summarised in tables with accompanying narrative summaries that included a description of the included variables, statistical methods and performance measures (eg, sensitivity, specificity and C-statistic (a value between 0.7 and 0.8 and >0.8 indicated good and excellent discrimination, respectively; and values <0.7 were considered weak[20]), where applicable. All analyses were conducted using Microsoft Excel V.2010 (Microsoft Corporation, Redmond, Washington, USA).

## Patient and public involvement

Patients and the public were not involved in the design or conduct of this systematic review.

## RESULTS

### Study flow

Figure 1 summarises the process of identifying and selecting relevant literature. Of the 6355 citations identified, 51 studies investigating 24 unique RAMs met the inclusion criteria. The majority of the articles were excluded primarily for not using a RAM for predicting the risk of developing VTE, having no useable or relevant outcome data or an inappropriate study design (eg, derivation study, reviews, commentaries or editorials). A

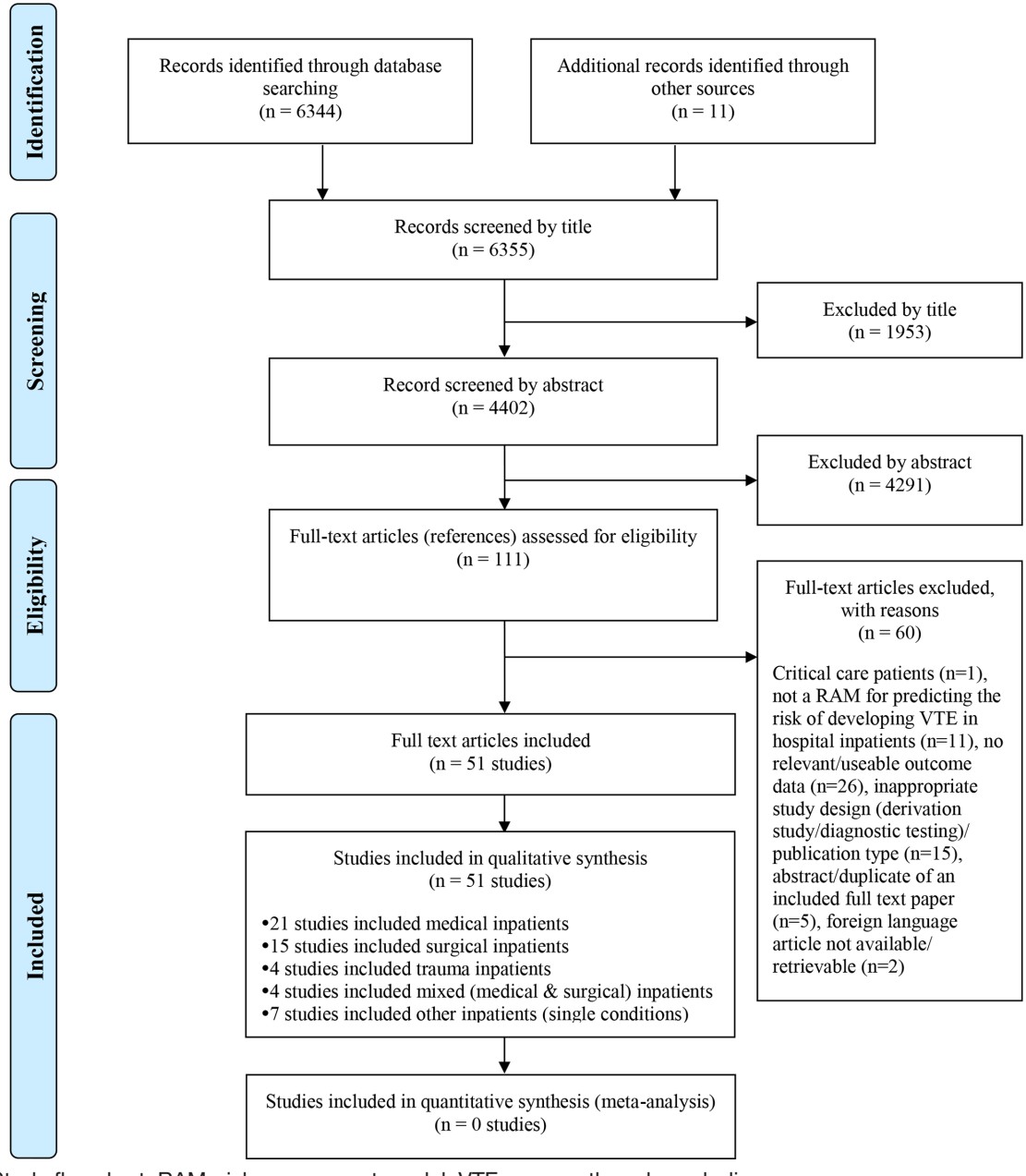

**Figure 1** Study flowchart. RAM, risk assessment model; VTE, venous thromboembolism.

**Table 1** Study and population characteristics

| Author, year | Country | Design | Single/ Multicentre | Sample size | Population | Mean age (years) | Male | VTE prophylaxis | RAMs | Target condition (risk period) | Incidence | Validation methodology |
|---|---|---|---|---|---|---|---|---|---|---|---|---|
| Autar, 2003[22] | UK | P,CS | Single | 148 | Hospitalised patients from orthopaedic, medical and surgical specialties | NR | NR | 50% | ▲ Novel (Autar, 2003) | DVT, not defined (90 days) | 18.9% | External |
| Rogers et al, 2007[56] | USA | P,CS | Multi | 91 308 | Hospitalised surgical patients (undergoing vascular and general surgery) | NR | NR | NR | ▲ Novel (Rogers et al, 2007) | VTE (30 days) | 0.6% | Internal: split (half) |
| Abdel-Razeq et al, 2010[21] | Jordan | P,CS | Single | 606 | Hospitalised (>24 hours) cancer patients aged ≥18 years | 51 | 51% | 55% | ▲ Caprini (modified) | VTE, symptomatic (60 days) | 3.5% | External |
| Bahl et al, 2010[23] | USA | R,CS | Multi | 8216 | Hospitalised surgical patients (undergoing general, vascular and urologic surgery) | NR | NR | NR | ▲ Caprini | VTE (30 days) | 1.4% | External |
| Barbar et al, 2010[24] | Italy | P,CS | Single | 1180 | Hospitalised medical patients | NR | 47% | 16% | ▲ Padua | VTE, symptomatic (90 days) | 3.1% | External |
| Rothberg et al, 2011[58] | USA | R,CS | Multi | 48 540 | Hospitalised (≥3 days) medical patients aged ≥18 years | NR | NR | 30% | ▲ Novel (Rothberg et al, 2011) | VTE, hospital associated (NR) | 0.5% | Internal: split (20%) |
| Woller et al, 2011[69] | USA | R,CS | Multi | 46 856 | Hospitalised medical patients aged ≥18 years | 61 | 46% | NR | ▲ Intermountain ▲ Kucher | VTE, defined by ICD-9 codes (90 days) | 4.5% | Internal: split (25%) |
| Pannucci et al, 2012[53] | USA and Canada | R,CS | Multi | 5761 | Hospitalised (>2 days) patients with a burn injury aged ≥18 years | 46 | 69% | NR | ▲ Novel (Panunucci et al, 2012) | VTE, not defined (NR) | 1.0% | Internal: split (25%) |
| Rogers et al, 2012[55] | USA | R,CS | Multi | 234 032 | Hospitalised trauma patients | NR | NR | NR | ▲ TESS | VTE (NR) | NR | Internal: split |
| Bilimoria et al, 2013[25] | USA | R,CS | Multi | 88 053 | Hospitalised surgical patients (undergoing colorectal surgery) | NR | NR | NR | ▲ ACS NSQIP—Colon specific ▲ ACS NSQIP—Universal | DVT, not defined (30 days) | 2.3% | External: split (by year) |
| Hegsted et al, 2013[39] | USA | R,CS | Single | 2281 | Hospitalised (≥2 days) trauma patients aged ≥13 years | 45 | 70% | NR | ▲ RAP | DVT, not defined or PE (NR) | ▲ DVT: 10.5% ▲ PE: 1.5% | External |
| Vardi et al, 2013[64] | Israel | P,CS | Single | 1080 | Hospitalised (≥2 days) sepsis patients aged >18 years | 75 | 52% | 18% | ▲ Padua | VTE, hospital associated (NR) | 1.3% | External |
| Ho et al, 2014[41] | Australia | R,CS | Single | 357 | Hospitalised major trauma patients | NR | 75% | NR | ▲ TESS | VTE, symptomatic (NR) | 20.7% | External |
| Liu et al, 2014[44] | China | P,CS | Single | 287 | Hospitalised acute stroke patients aged >18 years | NR | 63% | 22% | ▲ Post-stroke DVT prediction system | DVT (14±3 days) | 10.5% | Internal: split (33%) |

Continued

**Table 1** Continued

| Author, year | Country | Design | Single/Multicentre | Sample size | Population | Mean age (years) | Male | VTE prophylaxis | RAMs | Target condition (risk period) | Incidence | Validation methodology |
|---|---|---|---|---|---|---|---|---|---|---|---|---|
| Mahan et al, 2014[47] | USA | CC | Multi | 417 | Hospitalised (≥3 days) medical patients aged ≥18 years | NR | 49% | NR | ▲ IMPROVE (7-factor) | VTE, hospital associated (92 days) | NA | External |
| Nendaz et al, 2014[51] | Switzerland | P,CS | Multi | 1478 | Hospitalised (>24 hours) medical patients aged ≥18 years | 65 | 53% | 57% | ▲ Geneva ▲ Padua | VTE, symptomatic including PE or DVT (90 days) | 2.0% | External |
| Pannucci et al, 2014[52] | USA | P,CS | Multi | 3576 | Hospitalised surgical patients aged ≥18 years | NR | NR | 66% | ▲ Novel (Panunucci et al, 2014) | VTE (90 days) | 1.4% | Internal: split (35%) |
| Rosenberg et al, 2014[57] | USA | CC | Multi | 19 217 | Hospitalised (≥3 days) medical patients aged ≥18 years | NR | 47% | 43% | ▲ IMPROVE (7-factor) | VTE, defined by ICD-9 codes (90 days) | NA | External |
| Zhou et al, 2014[71] | China | CC | Single | 998 | Hospitalised (≥2 days) medical patients aged >18 years | NR | 58% | 15% | ▲ Caprini ▲ Padua | VTE, defined by ICD-10 codes (NR) | NA | External |
| Hewes et al, 2015[40] | USA | R,CS | Single | 70 | Hospitalised cancer patients (undergoing oesophagectomy) | NR | 83% | 96% | ▲ Caprini (modified) | VTE (60 days) | 14.3% | External |
| de Bastos et al, 2016[32] | Brazil | P,CS | Single | 11 091 | Hospitalised medical patients aged >18 years | 50 | 61% | 0% | ▲ Caprini | VTE, symptomatic (NR) | 0.3% | External |
| Grant et al, 2016[36] | USA | R,CS | Multi | 63 548 | Hospitalised (≥2 days) medical patients aged ≥18 years | 66 | 45% | 61% | ▲ Caprini | VTE, hospital associated (90 days) | 1.1% | External |
| Greene et al, 2016[37] | USA | R,CS | Multi | 63 548 | Acutely ill, hospitalised (≥2days) medical patients aged ≥18 years | 66 | 45% | 61% | ▲ IMPROVE (4-factor) ▲ Intermountain ▲ Kucher ▲ Padua | VTE, hospital associated (90 days) | 1.1% | External |
| Hachey et al, 2016[38] | USA | R,CS | Single | 232 | Hospitalised surgical patients (undergoing segmenectomy, lobectomy or pneumonectomy for lung cancer) | NR | 43% | 92% | ▲ Caprini | VTE (60 days) | 5.2% | External |
| Lui et al, 2016[45] | China | CC | Single | 640 | Hospitalised (>2 days) medical patients aged ≥18 years | NR | 52% | NR | ▲ Caprini ▲ Padua | VTE (NR) | N/A | External |
| Lobastov et al, 2016[46] | Russia | R,CS* | Multi | 140 | Hospitalised high-risk emergency surgery patients (undergoing general and neurosurgery) | 69 | 49% | 100% | ▲ Caprini | DVT or PE, new (NR) | 27.9% | External |
| Shaikh et al, 2016[59] | USA | R,CS | Multi | 1598 | Hospitalised surgical patients (undergoing plastic surgery) | 50 | 19% | 34% | ▲ Caprini | VTE, not defined (30 days) | 1.5% | External |

Continued

**Table 1** Continued

| Author, year | Country | Design | Single/ Multicentre | Sample size | Population | Mean age (years) | Male | VTE prophylaxis | RAMs | Target condition (risk period) | Incidence | Validation methodology |
|---|---|---|---|---|---|---|---|---|---|---|---|---|
| Elias et al, 2017[34] | USA | R,CS | Single | 30726 | Hospitalised (>2 days) medical and surgical patients | NR | 44% | 21% | ▲ Padua (automated) | VTE, defined by ICD-9 codes (NR) | 0.8% | External |
| Frankel et al, 2017 (abstract)[35] | USA | CC | NR | 149 | Hospitalised surgical patients aged ≥18 years (undergoing robotic-assisted laparoscopic prostatectomy) | NR | NR | NR | ▲ Caprini | VTE, not defined (90 days) | NA | External |
| Krasnow et al, 2017 (abstract)[43] | USA | R,CS | Multi | 1 099 093 | Hospitalised surgical patients (major urological cancer surgery) | NR | NR | NR | ▲ Caprini | VTE, symptomatic (90 days) | 1.2% | External |
| Patell et al, 2017[54] | USA | R,CS | Single | 2780 | Hospitalised (>24 hours) cancer patients aged >18 years | 62 (median) | 56% | 65% | ▲ Khorana | VTE, defined by ICD-9 codes (NR) | 3.8% | External |
| Winoker et al, 2017[68] | USA | R,CS | Multi | 300 | Hospitalised surgical patients (undergoing urological surgery using robot-assisted partial nephrectomy) | 61 (median) | 62% | NR | ▲ ACS NSQIP–Universal | VTE, not defined (NR) | 0.3% | External |
| Blondon et al, 2018[28] | Switzerland | P,CS | Multi | 1478 | Hospitalised (>24 hour) medical patients aged ≥18 years | 65 | 53% | 59% | ▲ IMPROVE (7-factor) ▲ Geneva † ▲ Padua † | VTE, symptomatic including PE or DVT (90 days) | 2.0% | External |
| Chen et al, 2018[30] | China | CC | Single | 390 | Hospitalised (>2 days) patients aged ≥18 years with and without DVT | NR | 51% | 41% | ▲ Caprini ▲ Padua | DVT (NR) | NA | External |
| Dornbus et al, 2018 (abstract)[33] | USA | R,CS | NR | 2830 | Hospitalised surgical patients (undergoing neurosurgery) | NR | NR | NR | ▲ Caprini | VTE, not defined (NR) | NR | External |
| Vaziri et al, 2018[65] | USA | R,CS | Single | 1006 | Hospitalised surgical patients (undergoing neurosurgery) | NR | 46% | NR | ▲ ACS NSQIP- Universal | VTE, not defined (NR) | 1.3% | External |
| Vincentelli et al, 2018[66] | Italy | CC | Multi | 1215 | Acutely ill, hospitalised medical patients aged >18 years | NR | 44% | NR | ▲ Chopard ▲ Kucher ▲ Padua | VTE (NR) | NA | External |
| Zhou et al, 2018[70] | China | CC | Single | 1804 | Hospitalised (≥2 days) medical patients aged >18 years | NR | 59% | 5% | ▲ Caprini ▲ Padua | VTE, defined by ICD-10 codes (NR) | NA | External |
| Blondon et al, 2019a[26] | Italy | R,CS* | Single | 1180 | Hospitalised medical patients | 72 | 47% | 20% | ▲ Geneva (simplified) | VTE, symptomatic (90 days) | 3.1% | External |
| Blondon et al, 2019b (abstract)[27] | Switzerland | R,CS * | Multi | 991 | Hospitalised elderly medical patients | 75 | 55% | NR | ▲ Geneva (simplified) ▲ IMPROVE (NR) ▲ Padua | VTE, symptomatic (NR) | 15.0% | External |

Continued

**Table 1** Continued

| Author, year | Country | Design | Single/Multicentre | Sample size | Population | Mean age (years) | Male | VTE prophylaxis | RAMs | Target condition (risk period) | Incidence | Validation methodology |
|---|---|---|---|---|---|---|---|---|---|---|---|---|
| Cobben et al, 2019[31] | Netherlands | CC | Multi | 556 | Hospitalised (>24 hours) medical patients | NR | 52% | NR | ▲ Caprini<br>▲ Geneva<br>▲ IMPROVE (4-factor)<br>▲ IMPROVE (7-factor)<br>▲ Intermountain<br>▲ Kucher<br>▲ Lecumberri<br>▲ NAVAL<br>▲ NICE Guideline<br>▲ Padua<br>▲ PRETEMED guideline<br>▲ Zakai et al (model 2) | VTE (NR) | NA | External |
| Tachino et al, 2019[62] | Japan | R,CS | Multi | 859 | Hospitalised (>24 hours) trauma patients aged ≥18 years | NR | 64% | NR | ▲ RAP<br>▲ Quick RAP | VTE (NR) | 3.0% | External (RAP)/internal (qRAP) |
| Tian et al, 2019[63] | China | R,CS | Single | 533 | Hospitalised surgical patients (undergoing thoracic surgery) | 53 | 53% | 0% | ▲ Caprini<br>▲ Khorana<br>▲ Padua<br>▲ Novel (Rogers et al, 2007) | VTE (NR) | 8.4% | External |
| Bo et al, 2020[29] | China | P,CS | Multi | 24524 | Hospitalised (≥2 days) patients from medical and surgical specialties aged ≥18 years | 57 | 57 | NR | ▲ Caprini | DVT (NR) | 0.9% | External |
| Hu et al, 2020[42] | China | CC | Single | 442 | Hospitalised (≥2 days) cancer patients aged ≥18 years | NR | 62 | 3.8 | ▲ Caprini<br>▲ Khorana | VTE, defined by ICD-10 codes (NR) | NA | External |
| Mlaver et al, 2020[48] | USA | CC | Single | 189 | Hospitalised surgical patients (undergoing hepatobiliary, colorectal, plastic, endocrine, transplant or general surgery) | NR | NR | NR | ▲ Caprini<br>▲ Padua | VTE, not defined (NR) | NA | External |
| Moumneh et al, 2020[49] | France | R,CS* | Multi | 14660 | Acutely ill, hospitalised (≥2 days) medical patients aged ≥40 years | 73 | 50 | 46.1 | ▲ Caprini<br>▲ Padua<br>▲ IMPROVE (7 factor) | VTE, symptomatic including PE or DVT (90 days) | 1.8% | External |
| Nafee et al, 2020[50] | 35 countries | R,CS* | Multi | 6459 | Hospitalised medical patients | 76 | 45 | 100 | ▲ IMPROVE (NR)<br>▲ Novel (Nafee et al, 2020a)<br>▲ Novel (Nafee et al, 2020b) | VTE (77 days) | 6.3% | External |

Continued

**Table 1** Continued

| Author, year | Country | Design | Single/Multicentre | Sample size | Population | Mean age (years) | Male | VTE prophylaxis | RAMs | Target condition (risk period) | Incidence | Validation methodology |
|---|---|---|---|---|---|---|---|---|---|---|---|---|
| Shang et al, 2020[60] | China | CC | Single | 2878 | Hospitalised (≥2 days) cancer patients aged ≥18 years | 56 | 47 | NR | ▲ Caprini (2009) ▲ Caprini (2013) | VTE, (NR) | NA | External |
| Shen et al, 2020[61] | China | CC | Single | 148 | Hospitalised (≥2 days) medical patients aged ≥18 years | NR | NR | 0 | ▲ Novel (Shen et al, 2020) | VTE, not defined (NR) | NA | Internal: split (by time, months) |
| Wang et al, 2020[67] | China | CC | Single | 1579 | Hospitalised (≥3 days) medical patients aged ≥18 years | 53 | 57 | NR | ▲ Padua | VTE, (NR) | NA | Internal: split (by year, months) |

*Prospective cohort study with retrospective analysis, thus classified as retrospective cohort study.
†Data overlap with Nendaz et al.[51]

ACS NSQIP, American College of Surgeons National Surgical Quality Improvement Program; CC, case-control; CS, cohort study; DVT, deep vein thrombosis; NA, not applicable; NR, not reported; P, prospective; PE, pulmonary embolism; R, retrospective; RAMs, risk assessment models; RAP, Risk Assessment Profile; TESS, Trauma Embolic Scoring System; VTE, venous thromboembolism.

full list of excluded studies with reasons for exclusion is provided in online supplemental appendix S2.

## Study and patient characteristics

The design and participant characteristics of the 51 included studies[21–71] are summarised in table 1. All studies were published between 2003 and 2020 and were undertaken in North America (n=24),[23 25 33–40 43 47 48 52–59 65 68 69] Asia (n=13),[29 30 42 44–46 60–63 67 70 71] Europe (n=9),[22 24 26–28 31 49 51 66] the Middle-East (n=2),[21 64] South America (n=1),[32] Australia (n=1)[41] and one study was intercontinental.[50] Sample sizes ranged from 70[40] to 1 099 093[43] patients in 37 observational cohort studies (11 prospective[21 22 24 28 29 32 44 51 52 56 64] (5 of which were multicentre) and 26 retrospective[23 25–27 33 34 36–41 43 46 49 50 53–55 58 59 62 63 65 68 69] (16 of which were multicentre) in design). Sample sizes in 14 case–control studies[30 31 35 42 45 47 48 57 60 61 66 67 70 71] (4 of which were multicentre) ranged from 148[61] to 19 217[57] patients.

The vast majority of studies evaluated VTE risk assessment in hospital inpatients who required medical care (n=21),[24 26–28 31 32 36 37 45 47 49–51 57 58 61 66 67 69–71] were undergoing surgery (n=15)[23 25 33 35 38 40 43 46 48 52 56 59 63 65 68] or were a mixed medical and surgical cohort (n=4).[22 29 30 34] The remaining studies focused on patients receiving care for trauma (n=4),[39 41 55 62] cancer (n=4),[21 42 54 60] stroke (n=1),[44] burn injuries (n=1)[53] and sepsis (n=1).[64] The mean age ranged from 45 years[39] to 76 years[50] (not reported in 29 studies)[22–25 30 31 33–35 38 40–45 47 48 52 55–58 61 62 65 66 70 71] and the proportion of female subjects ranged from 17%[40] to 81%[59] (not reported in 12 studies).[22 23 25 33 35 43 48 52 55 56 58 61]

## VTE definition and case ascertainment

The majority of studies (n=37)[21 23 24 26–32 36–38 40–47 49–52 55–58 60 62–64 66 67 70 71] defined the VTE endpoint (DVT and or PE) as being objectively confirmed. Of the remainder, 3 studies[34 54 69] had no objective confirmation of VTE and 11 studies[22 25 33 35 39 48 53 59 61 65 68] did not report the methods for diagnosis confirmation. In terms of VTE risk period, half of the studies (n=23)[21–26 28 35–38 40 43 44 47 49–52 56 57 59 69] used the RAMs to predict the occurrence of VTE within 3 months of the index hospitalisation. The remaining studies did not report the VTE risk period. The reported incidence of VTE ranged widely from 0.3%[32 68] to 27.9%,[46] depending on definition, study design and study participants (eg, medical, surgical or trauma).

## RAMs

The studies included in this review evaluated 24 validated unique RAMs. The most widely evaluated models were the Caprini RAM (22 studies),[21 23 29–33 35 36 38 40 42 43 45 46 48 49 59 60 63 70 71] Padua prediction score (16 studies),[24 27 28 30 31 34 37 45 48 49 63 64 66 67 70 71] IMPROVE models (8 studies),[27 28 31 37 47 49 50 57] the Geneva risk score (4 studies)[26–28 31] and the Kucher score (4 studies).[31 37 66 69] A summary of their associated characteristics and composite clinical variables is provided in online supplemental appendix S3.

**Table 2** Summary of each study's risk of bias and applicability concern using the PROBAST (Prediction model Risk Of Bias ASsessment Tool) tool—review authors' judgements

| Author, year | Risk of bias | | | | Concern regarding applicability | | | Overall | |
|---|---|---|---|---|---|---|---|---|---|
| | 1. Participant selection | 2. Predictors | 3. Outcome | 4. Analysis | 1. Participant selection | 2. Predictors | 3. Outcomes | Risk of bias | Applicability |
| Abdel-Razeq et al, 2010[21] | High | High | High | High | High | High | High | High | High |
| Autar, 2003[22] | High | High | High | High | High | High | High | High | High |
| Bahl et al, 2010[23] | High | High | High | High | Unclear | Unclear | Unclear | High | Unclear |
| Barbar et al, 2010[24] | Low | Unclear | Unclear | High | Low | Unclear | Unclear | High | Unclear |
| Bilimoria et al, 2013[25] | Low | Low | Low | High | Low | Low | Low | High | Low |
| Blondon et al, 2019a[26] | Low | Unclear | High | High | Low | Low | Low | High | Low |
| Blondon et al, 2019b (abstract)[27] | Unclear | Unclear | Unclear | Unclear | Unclear | Unclear | Unclear | Unclear | Unclear |
| Blondon et al, 2018[28] | Low | Unclear | Unclear | High | Unclear | Low | Unclear | High | Unclear |
| Bo et al, 2020[29] | Low | Unclear | Unclear | Unclear | High | Low | Low | Unclear | High |
| Chen et al, 2018[30] | High | High | High | High | Unclear | High | High | High | High |
| Cobben et al, 2019[31] | Unclear | Unclear | High | High | Unclear | Low | Unclear | High | Unclear |
| de Bastos et al, 2016[32] | High | Low | High | High | High | Low | Low | High | High |
| Dornbus et al, 2018 (abstract)[33] | High | Unclear | High | Unclear | Unclear | Unclear | Unclear | High | Unclear |
| Elias et al, 2017[34] | High | Unclear | High | High | Low | Low | High | High | High |
| Frankel et al, 2017 (abstract)[35] | High | Unclear | Unclear | High | High | Unclear | Unclear | High | High |
| Grant et al, 2016[36] | High | Unclear | Unclear | Unclear | Low | Low | Low | High | Low |
| Greene et al, 2016[37] | Unclear | Unclear | Unclear | Unclear | Low | Low | Low | Unclear | Low |
| Hachey et al, 2016[38] | High | Unclear | Unclear | High | High | Low | High | High | High |

Continued

**Table 2** Continued

| Author, year | Risk of bias | | | | Concern regarding applicability | | | Overall | |
|---|---|---|---|---|---|---|---|---|---|
| | 1. Participant selection | 2. Predictors | 3. Outcome | 4. Analysis | 1. Participant selection | 2. Predictors | 3. Outcomes | Risk of bias | Applicability |
| Hegsted et al, 2013[39] | High | Unclear | High | High | High | Low | Unclear | High | High |
| Hewes et al, 2015[40] | High | Unclear | Unclear | High | High | Unclear | Low | High | High |
| Ho et al, 2014[41] | Unclear | Unclear | Unclear | High | High | Unclear | Unclear | High | High |
| Hu et al, 2020[42] | Unclear | Unclear | Unclear | Unclear | High | Unclear | Unclear | Unclear | High |
| Krasnow et al, 2017 (abstract)[43] | Unclear | Unclear | Unclear | Unclear | High | Unclear | Unclear | Unclear | High |
| Liu et al, 2014[44] | Low | Low | Unclear | Unclear | High | High | High | Unclear | High |
| Liu et al, 2016[45] | High | Unclear | High | High | High | Low | Low | High | High |
| Lobastov et al, 2016[46] | Unclear | Unclear | Unclear | High | High | Low | High | High | High |
| Mahan et al, 2014[47] | Low | Unclear | Unclear | Unclear | High | Low | Unclear | Unclear | High |
| Mlaver et al, 2020[48] | Unclear | Unclear | Unclear | Unclear | High | Unclear | Unclear | Unclear | High |
| Moumneh et al, 2020[49] | High | Unclear | Unclear | Low | High | Low | Low | High | High |
| Nafee et al, 2020[50] | Unclear | Low | Low | Low | Unclear | Low | Low | Unclear | Unclear |
| Nendaz et al, 2014[51] | Low | Unclear | Low | High | Low | Unclear | Low | High | Unclear |
| Pannucci et al, 2012[53] | High | Unclear | Unclear | High | High | High | Unclear | High | High |
| Pannucci et al, 2014[52] | Low | Unclear | High | High | High | Low | Low | High | High |
| Patell et al, 2017[54] | High | Unclear | Unclear | High | High | Unclear | Unclear | High | High |
| Rogers et al, 2007[56] | Unclear | Unclear | Unclear | High | Low | Unclear | Unclear | High | Unclear |
| Rogers et al, 2012[55] | High | High | Unclear | High | High | High | Unclear | High | High |
| Rosenberg et al, 2014[57] | Low | Unclear | Unclear | Unclear | Unclear | Unclear | Unclear | Unclear | Unclear |

**Table 2** Continued

| Author, year | Risk of bias | | | | Concern regarding applicability | | | Overall | Overall |
|---|---|---|---|---|---|---|---|---|---|
| | 1. Participant selection | 2. Predictors | 3. Outcome | 4. Analysis | 1. Participant selection | 2. Predictors | 3. Outcomes | Risk of bias | Applicability |
| Rothberg et al, 2011[58] | High | Unclear | Unclear | High | Low | Unclear | Unclear | High | Unclear |
| Shaikh et al, 2016[59] | High | Unclear | High | High | High | Unclear | High | High | High |
| Shang et al, 2020[60] | Low | Unclear | Unclear | Unclear | High | Unclear | Unclear | Unclear | High |
| Shen et al, 2020[61] | Unclear | High | Unclear | Unclear | High | Unclear | Unclear | High | High |
| Tachino et al, 2019[62] | High | Unclear | Unclear | High | High | Unclear | Unclear | High | High |
| Tian et al, 2019[63] | High | Unclear | High | High | High | High | High | High | High |
| Vardi et al, 2013[64] | Unclear | Low | Low | High | High | Low | Low | High | High |
| Vaziri et al, 2018[65] | Unclear | Unclear | Unclear | High | High | Unclear | Unclear | High | High |
| Vincentelli et al, 2018[66] | High | Low | Unclear | High | High | Low | Unclear | High | High |
| Wang et al, 2020[67] | Low | Unclear | Unclear | Unclear | High | Unclear | Unclear | Unclear | High |
| Winoker et al, 2017[68] | High | Unclear | Unclear | High | High | High | High | High | High |
| Woller et al, 2011[69] | High | High | Unclear | High | Unclear | Unclear | Unclear | High | Unclear |
| Zhou et al, 2014[71] | Unclear | Unclear | Unclear | High | High | Unclear | Unclear | High | High |
| Zhou et al, 2018[70] | Low | High | High | High | High | Unclear | Unclear | High | High |

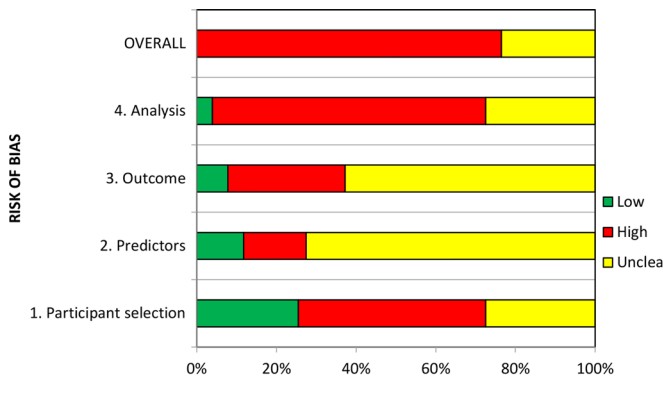

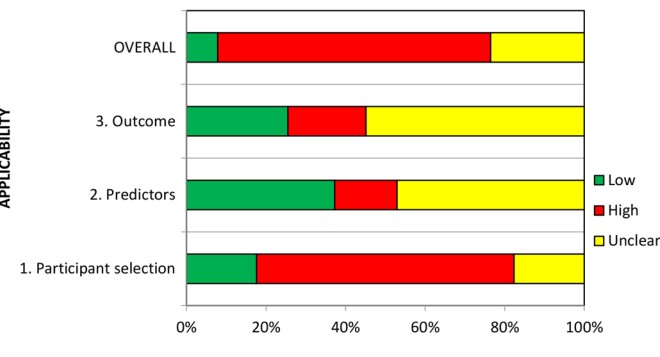

**Figure 2** PROBAST (Prediction model Risk Of Bias ASsessment Tool) assessment summary graph—review authors' judgements.

## Statistical methods

Statistical methods varied significantly between studies. Most studies reported the discrimination of the RAMs using a combination of the C-statistic and sensitivity or specificity. A minority reported calibration measures, such as the Hosmer-Lemeshow test.[23 40 41 50]

## Risk of bias and applicability assessment

The overall methodological quality of the 51 included studies[21–71] is summarised in table 2 and figure 2. The methodological quality of the included studies was variable, with most studies having high or unclear risk of bias in at least one item of the PROBAST tool. The main sources of potential bias were related to the following domains:

1. Patient selection factors, such as retrospective data collection, incomplete patient enrolment or unclear criteria for patients receiving VTE prophylaxis.
2. Predictor and outcome bias arising from inappropriate inclusion of predictors within RAMs, unclear methods of outcome definition, low event rates and missing predictor or outcome data.
3. Analysis factors, such as small sample sizes, inappropriate handling of missing data and failure in reporting relevant performance measures such as calibration.

Assessment of applicability to the review question led to the majority of studies being classed either as high (n=35)[21 22 29 30 32 34 35 38–49 52–55 59–68 70 71] or unclear (n=12)[23 24 27 28 31 33 50 51 56–58 69] risk of inapplicability. These assessments were generally related to patient selection (highly selected study populations, eg, single pathologies, single site settings), predictors (inconsistency in definition, assessment or timing of predictors) and outcome determination.

### Predictive performance of VTE RAMs (summary of results)

As there were a reasonable number of studies to compare, a summary of the C-statistics for studies involving medical, surgical and trauma patients respectively is presented in figure 3a–c, with the results grouped by RAM. Results of other hospital inpatients are presented in online supplemental appendix S4. C-statistics varied markedly between these studies and between models, with no RAM performing obviously better than other models. In studies evaluating a single model, C-statistics[20] were sometimes weak (<0.7; 10 studies with 17 data points), often good (0.7–0.8; 17 studies with 20 data points) and a few were excellent (>0.8; 5 studies with 5 data points). There was marked heterogeneity between multiple studies evaluating the same model. Studies evaluating multiple (more than 3) models[31 37] tended to report weak accuracy across all the models (C-statistic <0.7; 2 studies with 16 data points).

Table 3 shows the sensitivity and specificity at various thresholds for studies involving medical, surgical and trauma patients respectively, with the results grouped by RAM. Interpretation was again limited by marked heterogeneity, which was exacerbated when different thresholds were reported by different studies evaluating the same model. Model accuracy was generally poor, with high sensitivity usually reflecting a threshold effect, as evidenced by corresponding low specificity (and vice versa).

## DISCUSSION
### Summary of results

In this systematic review of 51 observational studies evaluating RAMs for predicting the risk of developing VTE in hospital inpatients, we found that VTE RAMs have generally weak predictive accuracy. The studies validating these models are heterogeneous and most have a high risk of bias. Lack of methodological clarity was common, leading to difficulty in assessing the applicability of the individual study results.

### Interpretation of results

We were unable to undertake meta-analysis or statistical examination of the causes of the observed heterogeneity. Potential sources of heterogeneity include variation in study design, the study population, how RAMs are implemented, outcome definition and measurement, and the use of thromboprophylaxis.

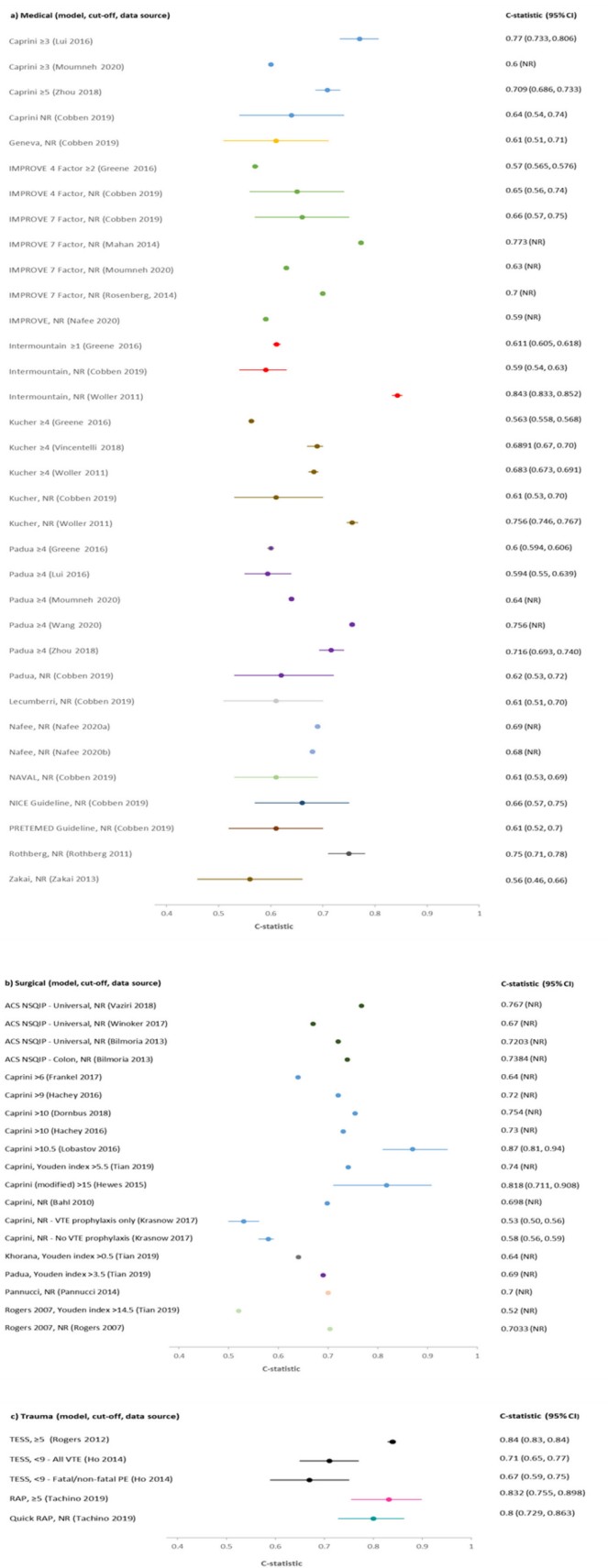

**Figure 3** C-statistics by model for studies involving (a) medical, (b) surgical and (c) trauma inpatients. ACS NSQIP, American College of Surgeons National Surgical Quality Improvement Program; CI, confidence interval; DVT, deep vein thrombosis; NR, not reported; PE, pulmonary embolism; RAP, Risk Assessment Profile; TESS, Trauma Embolic Scoring System; VTE, venous thromboembolism.

**Table 3** Sensitivity and specificity for studies involving medical, surgical and trauma inpatients

| Risk assessment models | Threshold or cut-off | Endpoint | Data source | Sensitivity (95% CI) | Specificity (95% CI) |
|---|---|---|---|---|---|
| MEDICAL INPATIENTS | | | | | |
| Caprini (7 studies) | Risk score ≥3 | VTE | Lui et al, 2016[45] | 70.9% (NR) | 73.4% (NR) |
| | Risk score ≥3 | VTE | Moumneh et al, 2020[49] | 98.1% (95.6 to 99.4) | 7.5% (7.1 to 8.0) |
| | Risk score ≥3 | VTE | Zhou et al, 2014[71] | 82.3% (NR) | 60.4% (NR) |
| | Risk score ≥3 | VTE | Zhou et al, 2018[70] | 84.3% (NR) | 66.2% (NR) |
| | Risk score ≥5 | VTE | Zhou et al, 2018[70] | 57.1% (NR) | 24.6% (NR) |
| | Risk score ≥5 | VTE | Grant et al, 2016[36] | 69.7% (NR) | 50.28% (NR) |
| | Risk score ≥7 | VTE | Grant et al, 2016[36] | 42.69% (NR) | 74.71% (NR) |
| | Risk score ≥9 | VTE | Grant et al, 2016[36] | 18.51% (NR) | 89.03% (NR) |
| | NR* | VTE | de Bastos et al, 2016[32] | 86.5% (NR) | 47.0% (NR) |
| | NR | VTE | Cobben et al, 2019[31] | 88.6% (NR) | 21.4% (NR) |
| Chopard (1 study) | Risk score ≥3 | VTE | Vincentelli et al, 2018[66] | 64.2% (38.4 to 81.9) | 57.7% (63.9 to 79.4) |
| Geneva models (4 studies) | Risk score ≥3 | VTE | Blondon et al, 2018[28]; Nendaz et al, 2014[51] | All patients: 90.0% (73.5 to 97.9) | All patients: 35.3% (32.8 to 37.8) |
| | | | | No prophylaxis: 85% (NR) | No prophylaxis: NR |
| | NR | VTE | Cobben et al, 2019[31] | 75.0% (NR) | 34.1% (NR) |
| | Simplified model: Risk score ≥3 | VTE | Blondon et al, 2019a[26] | 95.0% (NR) | 44.0% (NR) |
| | Simplified model: NR | VTE | Blondon et al, 2019b (abstract)[27] | 86.4% (NR) | NR |
| IMPROVE models (4 studies) | 4-factor model: NR | VTE | Cobben et al, 2019[31] | 27.9% (NR) | 85.4% (NR) |
| | 7-factor model: Risk score ≥2 | VTE | Moumneh et al, 2020[49] | 73.8% (68.0 to 79.0) | 47.1% (46.3 to 47.9) |
| | 7-factor model: Risk score 2–3 | VTE | Blondon et al, 2018[28]; Nendaz et al, 2014[51] | All patients: 87% (NR) | All patients: NR |
| | | | | No prophylaxis: 85% (NR) | No prophylaxis: NR |
| | 7-factor model: Risk score ≥3 | VTE | Blondon et al, 2018[28]; Nendaz et al, 2014[51] | All patients: 73% (NR) | All patients: NR |
| | | | | No prophylaxis: 54% (NR) | No prophylaxis: NR |
| | 7-factor model: Risk score ≥4 | VTE | Moumneh et al, 2020[49] | 24.7% (19.6 to 30.4) | 85.5% (84.9 to 86.1) |
| | 7-factor model: NR | VTE | Cobben et al, 2019[31] | 63.3% (NR) | 70.7% (NR) |
| | NR | VTE | Blondon et al, 2019b (abstract)[27] | 57.6% (NR) | NR |
| Intermountain (1 study) | NR | VTE | Cobben et al, 2019[31] | 26.4% (NR) | 90.2% (NR) |
| Kucher (2 studies) | Risk Score ≥4 | VTE | Vincentelli et al, 2018[66] | 25.1% (17.0 to 55.1) | 92.9% (81.0 to 95.4) |
| | NR | VTE | Cobben et al, 2019[31] | 28.0% (NR) | 85.7% (NR) |
| Lecumberri (1 study) | NR | VTE | Cobben et al, 2019[31] | 61.6% (NR) | 46.3% (NR) |
| NAVAL (1 study) | NR | VTE | Cobben et al, 2019[31] | 19.0% (NR) | 92.7% (NR) |
| NICE Guidelines (1 study) | NR | VTE | Cobben et al, 2019[31] | 77.6% (NR) | 39.0% (NR) |
| Padua (10 studies) | Risk score ≥4 | VTE | Barbar et al, 2010[24] | 94.6% (NR) | 62.0% (NR) |

Continued

**Table 3** Continued

| Risk assessment models | Threshold or cut-off | Endpoint | Data source | Sensitivity (95% CI) | Specificity (95% CI) |
|---|---|---|---|---|---|
| | Risk score ≥4 | VTE | Blondon et al, 2018[28]; Nendaz, 2014[51] | All patients: 73.3% (54.1 to 87.7) | All patients: 51.9% (49.3 to 54.5) |
| | | | | No prophylaxis: 62% (NR) | No prophylaxis: NR |
| | Risk score ≥4 | VTE | Lui et al, 2016[45] | 23.4% (NR) | 85.6% (NR) |
| | Risk score ≥4 | VTE | Moumneh et al, 2020[49] | 91.6% (87.6 to 94.7) | 25.6% (24.9 to 26.3) |
| | Risk score ≥4 | VTE | Zhou et al, 2014[71] | 30.1% (NR) | 12.7% (NR) |
| | Risk score ≥4 | VTE | Zhou et al, 2018[70] | 49.1% (NR) | 16.2% (NR) |
| | Risk score ≥4 | VTE | Vincentelli et al, 2018[66] | 52.4% (38.4 to 81.9) | 72.3% (63.9 to 79.4) |
| | Risk score ≥4 | VTE | Wang et al, 2020[67] | 76.2% (NR) | 61.6% (NR) |
| | NR | VTE | Blondon et al, 2019b (abstract)[27] | 72.7% (NR) | NR |
| | NR | VTE | Cobben et al, 2019[31] | 61.8% (NR) | 48.8% (NR) |
| PRETEMED guidelines (1 study) | NR | VTE | Cobben et al, 2019[31] | 81.6% (NR) | 24.4% (NR) |
| Shen 2020 (1 study) | NR | VTE | Shen et al, 2020[61] | 77.8% (NR) | 84.7% (NR) |
| Zakai 2013 (1 study) | Model 2: NR | VTE | Cobben et al, 2019[31] | 63.8% (NR) | 31.7% (NR) |
| SURGICAL INPATIENTS | | | | | |
| Caprini (8 studies) | Risk score >5 | VTE | Hachey et al, 2016[38] | 100% (100 to 100) | 7.2% (4.1 to 11.0) |
| | Risk score ≥5 | VTE | Mlaver et al, 2020[48] | 88.9% (NR) | 32.7% (NR) |
| | Risk score >5 | VTE | Shaikh et al, 2016[59] | 70.8% (48.9 to 87.4) | 39.39% (37.0 to 41.9) |
| | Youden index >5.5 | VTE | Tian et al, 2019[63] | 76.0% (NR) | 64.0% (NR) |
| | Risk score >6 | VTE | Frankel et al, 2017 (abstract)[35] | 61.5% (NR) | 59.8% (NR) |
| | Risk score >6 | VTE | Shaikh et al, 2016[59] | 58.3% (36.6 to 77.9) | 60.1% (57.6 to 62.5) |
| | Risk score >7 | VTE | Hachey et al, 2016[38] | 100% (100 to 100) | 31.4% (25 to 37.3) |
| | Risk score >9 | VTE | Hachey et al, 2016[38] | 83.3% (58.3 to 100) | 60.5% (54.4 to 67.3) |
| | Risk score >9 | VTE | Shaikh et al, 2016[59] | 16.7% (NR) | 93.3% (NR) |
| | Risk score >10 | VTE | Hachey et al, 2016[38] | 75.0% (50 to 100) | 69.6% (64.6 to 76.4) |
| | Risk score >10 | VTE | Dornbus et al, 2018 (abstract)[33] | 78.9% (NR) | 60.9% (NR) |
| | Risk score >10.5 | DVT or PE | Lobastov et al, 2016[46] | 95.0% (NR) | 73.0% (NR) |
| | Risk score >15 † | VTE | Hewes et al, 2015[40] | 100% (100 to 100) | 66.7% (55.0 to 78.3) |
| Khorana (1 study) | Youden index >0.5 | VTE | Tian et al, 2019[63] | 78.0% (NR) | 48.0% (NR) |
| Padua (2 studies) | Risk score ≥4 | VTE | Mlaver et al, 2020[48] | 61.1% (NR) | 47.4% (NR) |
| | Youden index >3.5 | VTE | Tian et al, 2019[63] | 36.0% (NR) | 93.0% (NR) |
| Rogers 2007 (1 study) | Youden index >14.5 | VTE | Tian et al, 2019[63] | 53.0% (NR) | 54.0% (NR) |
| TRAUMA PATIENTS | | | | | |
| RAP (2 studies) | Risk score ≥5 | VTE | Tachino et al, 2019[62] | 100% (86.8 to 100) | 37.9% (34.6 to 41.3) |
| | Risk score 5 to ≤14 | DVT or PE | Hegsted et al, 2013[39] | ► DVT: 82.0% (77 to 87) ► PE: 71.0% (55 to 86) | ► DVT: 57.0% (55 to 59) ► PE: 53.0% (51 to 56) |

**Table 3** Continued

| Risk assessment models | Threshold or cut-off | Endpoint | Data source | Sensitivity (95% CI) | Specificity (95% CI) |
|---|---|---|---|---|---|
| | Risk score >14 | DVT or PE | Hegsted et al, 2013[39] | ▶ DVT: 15.0% (11 to 20)<br>▶ PE: 12.0% (1 to 23) | ▶ DVT: 97.0% (97 to 98)<br>▶ PE: 96.0% (95 to 97) |
| TESS (2 studies) | Risk score ≥5 | VTE | Rogers et al, 2012[55] | 77.4% (NR) | 75.6% (NR) |
| | Risk score <9 | VTE | Ho et al, 2014[41] | ▶ All VTE: 97.0% (91 to 99) | ▶ All VTE: 27.0% (22 to 32) |
| | Risk score <9 | VTE | Ho et al, 2014[41] | ▶ Fatal and non-fatal PE: 97.0% (87 to 99) | ▶ Fatal and non-fatal PE: 24.0% (20 to 29) |
| | Risk score <9 | VTE | Ho et al, 2014[41] | ▶ Fatal PE only: 100% (81 to 100) | ▶ Fatal PE only: 20.0% (13 to 28) |

*Paper states 'moderate and high risk'.
†Modified Caprini model.
.DVT, deep vein thrombosis; NR, not reported; PE, pulmonary embolism; RAP, Risk Assessment Profile; TESS, Trauma Embolic Scoring System; VTE, venous thromboembolism.

The latter point warrants further attention. Thromboprophylaxis was employed in about half (n=25) of the studies,[21 22 24 26 28 30 34 36–38 40 42 44 46 49–52 54 57–59 64 70 71] with the proportion receiving thromboprophylaxis ranging from 3.8%[42] to 100%.[46 50] It was not employed in 3 studies,[32 61 63] and 23 studies[23 25 27 29 31 33 35 39 41 43 45 47 48 53 55 56 60 62 65–69] did not report on thromboprophylaxis use. The use of thromboprophylaxis may lead to underestimation of predictive accuracy if a given RAM were to predict VTE events that were subsequently prevented by thromboprophylaxis. Limited reporting of thromboprophylaxis use precludes further analysis of its impact on the performance of the RAMs.

### Comparison to the existing literature

The present review is the largest and most comprehensive systematic review in this field to date. It includes 18 recent studies[26–31 33 42 48–50 60–63 66 67 70] published since the completion of the previous systematic review.[10 12 13] These studies are consistent with the previous literature in that they report modest performance of the assessed RAMs, with limitations in methodology and reporting preventing further analysis. The conclusion of this review therefore concurs with previous systematic reviews: there is insufficient evidence to recommend one RAM over another.

### Strengths and limitations

This systematic review has a number of strengths. The review was conducted with robust methodology in accordance with the PRISMA statement and the protocol was registered with the PROSPERO register. Clinical experts were involved throughout as checkers and to assess the validity and applicability of research during the review. We reported descriptive statistics to provide insight into the limited evidence base applicable to the subject matter, and the scientific concerns regarding validity of the data. However, there are a number of potential weaknesses. Decisions on study relevance, information gathering and validity were unblinded and could potentially have been influenced by pre-formed opinions. However, masking is resource intensive with uncertain benefits. The studies of risk prediction were a combination of prospective cohorts and retrospective health database registries. Both have significant limitations. Retrospective studies of health database registries may have large numbers but may be limited by poor data quality and failure to accurately ascertain outcomes. Prospective cohorts may have better quality data but with smaller numbers lack statistical power. The included studies demonstrated high levels of heterogeneity so we were unable to undertake any meta-analysis.

### Implications for policy, practice and future research

Guidelines from the American College of Chest Physicians (ACCP)[72 73] and the UK National Institute for Health and Care Excellence (NICE)[10] suggest using a validated RAM to guide the decision on whether to prescribe thromboprophylaxis. This review identifies all relevant RAMs and their validation studies. The reported results are insufficient to recommend one RAM over another. A RAM with weak predictive accuracy may still be better than no RAM at all but it is unclear whether RAMs predict VTE risk better than unstructured clinical assessment. Further research is clearly needed but routine use of thromboprophylaxis may present an insurmountable barrier to generating accurate and precise estimates of the prognostic accuracy of RAMs. The evidence that thromboprophylaxis is effective means that it is unethical to withhold thromboprophylaxis when a significant risk of VTE is identified. This inevitably reduces the number of VTE events in any study and confounds the association

between risk factors and VTE events. Further studies of RAM accuracy will add little to our review unless they can address this issue.

Alternative approaches therefore need to be considered. Decision-analytic modelling can use existing data to explore the trade-off between the benefits and harms of thromboprophylaxis and identify key uncertainties for future primary research. The data presented in our review show how well RAMs predict VTE but do not tell us the threshold score on the RAM at which thromboprophylaxis should be given to maximise prevention of VTE and minimise harm from bleeding. This may be a more important determinant of RAM effectiveness than predictive accuracy for VTE. Le et al[74] suggested thromboprophylaxis is beneficial and cost-effective if a patient's VTE risk exceeds 1%. Further work to improve RAMs to help stratify the risk of VTE in different types of hospitalised patients could focus on using decision-analytic modelling to compare the effects, harms and costs of giving thromboprophylaxis to patients with varying risk of VTE. This would allow determination of the risk threshold at which thromboprophylaxis provides optimal overall benefit.

Findings from decision-analytic modelling would require validation through primary research. The limitations of undertaking accuracy studies in populations where thromboprophylaxis is routinely used mean that future research should focus on research that compares the effectiveness of different risk assessment approaches. Observational studies could draw on variation in practice to compare outcomes between different risk assessment methods. Alternatively, a controlled trial could compare risk assessment methods in low-risk patients where existing evidence (synthesised using decision-analytic modelling) suggests the benefits of thromboprophylaxis are uncertain.

## CONCLUSIONS
We identified a number of validated RAMs for potential risk stratification of hospitalised inpatients. The available evidence is insufficient to recommend one over another.

**Acknowledgements** The authors would like to thank all additional members of the core project group for NIHR HTA 127454 for input and commentary throughout the work. We are also indebted to Helen Shulver for assistance with logistics and administration.

**Contributors** AP coordinated the study. SG, DH, AP, XG, MH, BH and KW were responsible for conception, design and obtaining funding for the study. MC developed the search strategy, undertook searches and organised retrieval of papers. AP, KS, MT and SG were responsible for the acquisition, analysis and interpretation of data. SG, MT, DH, XG, MH, BH and KW helped interpret and provided a methodological, policy and clinical perspective on the data. AP, MT and SG were responsible for the drafting of this paper, although all authors provided comments on the drafts, read and approved the final version. AP is the guarantor for the paper.

**Funding** This study was funded by the United Kingdom National Institute for Health Research Health Technology Assessment Programme (project number 127454). The views expressed in this report are those of the authors and not necessarily those of the NIHR HTA Programme. Any errors are the responsibility of the authors. The funders had no role in the study design, in the collection, analysis and interpretation of data; in the writing of the manuscript; and in the decision to submit the manuscript for publication.

**Competing interests** None declared.

**Patient consent for publication** Not required.

**Provenance and peer review** Not commissioned; externally peer reviewed.

**Data availability statement** All data relevant to the study are included in the article or uploaded as supplementary information.

**ORCID iDs**
Abdullah Pandor http://orcid.org/0000-0003-2552-5260
Steve Goodacre http://orcid.org/0000-0003-0803-8444
Xavier L Griffin http://orcid.org/0000-0003-2976-7523
Kerstin de Wit http://orcid.org/0000-0003-2763-6474
Daniel Horner http://orcid.org/0000-0002-0400-2017

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
