## [Reviewer comments · BMJ Open]

ARTICLE DETAILS

TITLE (PROVISIONAL)	Risk assessment models for venous thromboembolism in hospitalised adult patients – a systematic review
AUTHORS	Pandor, Abdullah; Tonkins, Michael; Goodacre, Steve; Sworn, Katie; Clowes, Mark; Griffin, Xavier; Holland, Mark; Hunt, Beverley; de Wit, Kerstin; Horner, Daniel

VERSION 1 – REVIEW

REVIEWER	Khalafallah, Alhossain Specialist Care Australia, Medicine and Clinical Haematology
REVIEW RETURNED	16-Dec-2020

GENERAL COMMENTS	Review of “Risk assessment models for venous thromboembolism in hospitalised adult patients – a systematic review This sytemic review in general is valuable addition and considered good efforts to address this issue. I hope it will benefit future risk-assessment scores of VTE even it will not be of help in patients’ management due to the inconclusiveness of most of included studies. I have few comments that need to be addressed by the authors. In the abstract, the authors should cite the beginning of search date as they mentioned only the end at Sept 2019. Properly the authors should mention how the trials were excluded and how many trials in total were evaluated. I agree with the authors that due to the heterogeneity of populations and cohorts studied, it is difficult to draw any conclusion without making unsubstantiated assumption. The authors perhaps need to extend more discussion regarding the difference and analysis of RAM that were included in their analyses. I am not sure why Khorana score studies were not included if the systematic review as I expected to include all available studies about VTE. However, Khorana score or risk assessment was mainly for VTE-risk in oncology patients. I understand this might be a different topic, but it will be great if the authors complete their comprehensive review by including VTE-risk assessment also in specialised area e.g. oncology. The authors mentioned that studies with any language had been included, however, I’m not sure how these studies were translated, assessed and analysed if the languages other than English. Please add in the methods section how the authors dealt with this issue. Although it was mentioned that other non-English speaking countries were included in the analysis, it is unclear if the authors considered patients’ demographics and ethnicity when they included these studies in the meta-analysis.
--

	The authors should ensure that all variables had been studied and were measured in the same units and recorded at the same time in all included studies, if not, how the authors have addressed and adjusted the issue of different measurement units from study to study and country to another in their analyses. The authors had determined the quality of the included studies; however, it is unclear to what criteria they utilised. Perhaps a table showing the different studied trials against column of assessments will be a useful objective assessment. It will be great if the authors can discuss how future research can be improved with increasing influx of data on VTE prevention with different guidelines, policies and risk scores. Furthermore, how the research on VTE RAM can avoid bias as possible to improve study's validation and be more predictable of VTE events.
--	--

REVIEWER	Brækkan, Sigrid University of Tromsø
REVIEW RETURNED	22-Dec-2020

GENERAL COMMENTS	The authors have performed a systematic review of risk assessment models (RAMs) for venous thromboembolism in hospitalized adult patients. A protocol was developed a priori and appropriately registered in the Prospero database. Only RAMs developed for general inpatient populations were reviewed, and only primary validation studies were included. The review presented 44 studies evaluating 23 unique RAMs. The methods are well performed, and in accordance with the PRISMA guidelines for systematic reviews. The literature search is well described. It is a strength that only primary validation studies were included, as derivation studies are often overoptimistic with regards to performance measures. The tables give a thorough overview of the study characteristics and the performance of the various models. Meta-analytical approaches were not performed, which is reasonable considering the heterogeneity of the studies. I only have minor comments: In the abstract, the authors could consider to report the proportion of studies with weak, good and excellent performance respectively. The same could be done in the results section (page 18, first para). Page 9, first paragraph states "<0.5 as very weak". Should it not be 0.6? Figure 2 needs a more explanatory figure text (make it easier for the reader to understand it). The resolution of figure 3 was poor, so I could not review it properly.
---

REVIEWER	Campbell, Kristen University of Colorado, Pediatrics
REVIEW RETURNED	26-Jan-2021

GENERAL COMMENTS	This is a thorough and well thought out review on RAMs in patients with venous thromboembolism. Minor edits:
--

	Abstract: Abstract, line 26: 'through' September 2019 Abstract, lines 43-50: stay consistent with summarizing numbers as "n=xx" or "xx studies", but not both Page 17, lines 20-22: can you provide the number of studies who fell into each c-statistic grouping (i.e. <0.7, n=xx (%). Figure 2: can you provide a more detailed figure caption explaining each domain, so the reader can interpret the figure (at a basic level) without going back to the text? Figure 3: these are too low of quality to read – can you please send these in higher resolution, and then I will make my final comments on this section. Throughout: I'm used to seeing multivariable, not multi-variable General comments/edits: 1. I would like to hear a bit more about the existing overlapping existing reviews mentioned in the introduction on line 41 – a short summary of what they found would be useful here. 2. I agree with not performing a meta-analysis on this study cohort, primarily because the definition of the outcome and the included patient population differed so widely between studies. However, it would be of interest to see (descriptively) if certain characteristics were more common in studies with lower c-statistics, such as: higher author defined risk of bias, internal vs. external validation, broader age range of patients. (maybe others too, I am a statistical reviewer so don't know the full clinical implications of this, so open to other ideas). Or if you already have examined this, state in the methods/discussion. 3. Discussion: what are your specific recommendations do you have to design/implement a comprehensive study to determine the best RAM in this population? I appreciate your discussion on the limitations of the studies presented here, but a bit more detail on how to carve a path forward would be beneficial.
--	--

VERSION 1 – AUTHOR RESPONSE

Reviewer #1 This sytemic review in general is valuable addition and considered good efforts to address this issue. I hope it will benefit future risk-assessment scores of VTE even it will not be of help in patients' management due to the inconclusiveness of most of included studies. I have few comments that need to be addressed by the authors.	Thank you for your in-depth review and your comments on the work. In response to your suggestions and questions, we have addressed the following issues in the revised manuscript:
In the abstract, the authors should cite the beginning of search date as they mentioned only the end at Sept 2019.	Thank you for raising this point. The methods section in the Abstract has been revised as follows:

	‘A systematic search was performed across five electronic databases (including MEDLINE, EMBASE and the Cochrane Library) from inception to February 2021.’ Further details on the search strategy (including the varying data coverage of each electronic database) is provided in online supplementary Appendix S1.
Properly the authors should mention how the trials were excluded and how many trials in total were evaluated.	Thank you for raising this point. To reflect the total number of records identified by our searches, we have amended the abstract as follows: ‘Among 6355 records, we included 51 studies, comprising 24 unique validated RAMs.’ Moreover, we applied the study eligibility criteria outlined on p6 using the study selection process outlined on p7. The PRISMA flow chart (Figure 1) shows that 60 studies were excluded following full text review and outlined the reasons for exclusion.
I agree with the authors that due to the heterogeneity of populations and cohorts studied, it is difficult to draw any conclusion without making unsubstantiated assumption.	Thank you for this comment.
The authors perhaps need to extend more discussion regarding the difference and analysis of RAM that were included in their analyses.	We agree that there are some interesting differences between the RAMs but decided against discussing these differences in detail and relating them to our analysis because we felt that the uncertainty and heterogeneity in the primary data undermined any attempts to determine whether differences between the RAMs explained differences in the reported results for each RAM. We would be reluctant to engage in speculation on this issue but for completeness, we have provided additional

	information in online supplementary Appendix S3 on the variables included in the most widely used generic RAMs.
I am not sure why Khorana score studies were not included in the systematic review as I expected to include all available studies about VTE. However, Khorana score or risk assessment was mainly for VTE-risk in oncology patients. I understand this might be a different topic, but it will be great if the authors complete their comprehensive review by including VTE-risk assessment also in specialised area e.g. oncology.	Thank you for raising this point. As noted in the methods section (and our PROSPERO protocol), we excluded studies where patients required a special level of care e.g. adults with a single failing organ system or post-operative care or patients 'stepping down' from higher levels of care. These patient groups have VTE risk profiles that differ markedly from the general inpatient population (the group of interest in our systematic review). In addition, the Khorana score is widely used in predicting venous thromboembolic events in ambulatory cancer patients and has been comprehensively reviewed by Mulder et al. 2019. As such, Khorana score studies that are not undertaken in hospital inpatients are not directly relevant to the research question. References Mulder FI, et al The Khorana score for prediction of venous thromboembolism in cancer patients: a systematic review and meta-analysis. Haematologica. 2019 Jun;104(6):1277-1287
The authors mentioned that studies with any language had been included, however, I'm not sure how these studies were translated, assessed and analysed if the languages other than English. Please add in the methods section how the authors dealt with this issue.	Thank you for highlighting this issue. As noted in the methods section, no language restrictions were used in our searches or selection criteria. For transparency and to inform readers of the review of the availability of other possibly relevant studies, we were unable to retrieve two potentially relevant full-text publications as these were unavailable/ unobtainable. As these were also foreign language publications, we have clearly listed this as additional information in the table of excluded studies (see Appendix S2) and the PRISMA flow diagram (Figure 1).
Although it was mentioned that other non-English speaking countries were included in the analysis, it is unclear if the authors considered	Thank you for raising this point. As noted in the methods section, we were unable to perform a meta-analysis due to significant levels of heterogeneity between studies (the validity of

patients' demographics and ethnicity when they included these studies in the meta-analysis.	this approach was also supported by peer reviewers #2 and #3, see comments below).
The authors should ensure that all variables had been studied and were measured in the same units and recorded at the same time in all included studies, if not, how the authors have addressed and adjusted the issue of different measurement units from study to study and country to another in their analyses.	Thank you for raising this point. As noted in the methods section, a pre-specified narrative synthesis approach was undertaken. All standard descriptive data were extracted from the original papers (including statistical methods and standard performance measures such as reported sensitivity, specificity and C-statistic).
The authors had determined the quality of the included studies; however, it is unclear to what criteria they utilised. Perhaps a table showing the different studied trials against column of assessments will be a useful objective assessment.	Thank you for your suggestion. As noted in the methods section, the methodological quality of each included study was assessed using the PROBAST tool^{1,2} (see the methods section for details of the domains covered as well as how the overall risk of bias assessments were judged). This tool was published in 2019 and is now widely used to guide the critical appraisal of prediction model studies. In accordance with the suggestion of the PROBAST authors,^{1,2} we have presented the results of all assessments of risk of bias and applicability for each study (see Table 2) and also identified the key issues across all studies (see Figure 2). References 1. Moons KGM, Wolff RF, Riley RD, et al. PROBAST: A Tool to Assess Risk of Bias and Applicability of Prediction Model Studies: Explanation and Elaboration. Ann Intern Med 2019;170(1):W1-w33. 2. Wolff RF, Moons KGM, Riley RD, et al. PROBAST: A Tool to Assess the Risk of Bias and Applicability of Prediction Model Studies. Ann Intern Med 2019;170(1):51-58.
It will be great if the authors can discuss how future research can be improved with increasing influx of data on VTE prevention with different guidelines, policies and risk scores. Furthermore, how the research on VTE RAM can avoid bias as possible to improve study's	Thank you for your suggestion. We have amended the discussion section and have provided suggestions for future research.

validation and be more predictable of VTE events.	
Reviewer #2 The authors have performed a systematic review of risk assessment models (RAMs) for venous thromboembolism in hospitalized adult patients. A protocol was developed apriori and appropriately registered in the prospero database. Only RAMs developed for general inpatient populations were reviewed, and only primary validation studies were included. The review presented 44 studies evaluating 23 unique RAMs. The methods are well performed, and in accordance with the PRISMA guidelines for systematic reviews. The literature search is well described. It is a strength that only primary validation studies were included, as derivation studies are often overoptimistic with regards to performance measures. The tables give a thorough overview of the study characteristics and the performance of the various models. Meta-analytical approaches were not performed, which is reasonable considering the heterogeneity of the studies. I only have minor comments:	Thank you for your in-depth review, and your positive comments on the conduct, standard and writing of our work. We agree this topic is important, very relevant and timely.
In the abstract, the authors could consider to report the proportion of studies with weak, good and excellent performance respectively. The same could be done in the results section (page 18, first para).	Thank you for your suggestion. We have amended the abstract and results section as suggested and have reported the proportion of studies with weak, good and excellent performance respectively. The abstract has been amended as follow:

	'Across all models, C-statistics were often weak (<0.7), sometimes good (0.7-0.8) and a few were excellent (>0.8).' The results section has been amended as follows: 'C-statistics varied markedly between these studies and between models, with no RAM performing obviously better than other models. In studies evaluating a single model, C-statistics²⁰ were sometimes weak (<0.7; 10 studies with 17 data points), often good (0.7-0.8; 17 studies with 20 data points) and a few were excellent (>0.8; 5 studies with 5 data points). There was marked heterogeneity between multiple studies evaluating the same model. Studies evaluating multiple (more than 3) models^{31 37} tended to report weak accuracy across all the models (C-statistic<0.7; 2 studies with 16 data points).'
Page 9, first paragraph states "<0.5 as very weak". Should it not be 0.6?	Thank you for raising this point. For consistency and to avoid confusion we have amended the text in the data synthesis and analysis section as follows: '...statistical methods and performance measures (e.g. sensitivity, specificity and C-statistic [a value between 0.7 to 0.8 and >0.8 indicated good and excellent discrimination, respectively; and values <0.7 were considered weak²⁰),'
Figure 2 needs a more explanatory figure text (make it easier for the reader to understand it).	Thank you for highlighting this point. Figure 2 has been amended with errors corrected and additional explanatory text added.

The resolution of figure 3 was poor, so I could not review it properly.	Please accept our apologies. Figure 3 has been revised and uploaded (for higher resolution) in a PDF Format (as advised by the BMJ Editorial Production Assistant team).
Reviewer #3 This is a thorough and well thought out review on RAMs in patients with venous thromboembolism.	Thank you for your in-depth review, and your positive comments on the conduct, standard and writing of our work.
Minor edits: Abstract: Abstract, line 26: 'through' September 2019	Thank you for your comment. As suggested we have amended (and updated, as recommended by the Associate Editor) the abstract.
Abstract, lines 43-50: stay consistent with summarizing numbers as "n=xx" or "xx studies", but not both	Thank you for your suggestion. We have amended the abstract as follows: 'Among 6355 records, we included 51 studies, comprising 24 unique validated RAMs. The majority included hospital inpatients who required medical care (21 studies), were undergoing surgery (15 studies) or receiving care for trauma (4 studies).'
Page 17, lines 20-22: can you provide the number of studies who fell into each c-statistic grouping (i.e. <0.7, n=xx (%).	Thank you for your suggestion. We have amended the results section as follows: 'C-statistics varied markedly between these studies and between models, with no RAM performing obviously better than other models. In studies evaluating a single model, C-statistics²⁰ were sometimes weak (<0.7; 10 studies with 17 data points), often good (0.7-0.8; 17 studies with 20 data points) and a few were excellent (>0.8; 5 studies with 5 data points).

	There was marked heterogeneity between multiple studies evaluating the same model. Studies evaluating multiple (more than 3) models ^{31 37} tended to report weak accuracy across all the models (C-statistic<0.7; 2 studies with 16 data points).¹
Figure 2: can you provide a more detailed figure caption explaining each domain, so the reader can interpret the figure (at a basic level) without going back to the text?	Thank you for highlighting this point. Figure 2 has been amended with errors corrected and additional explanatory text added. In addition, Figure 2 has been presented in accordance to the guidance provided by the PROBAST authors,^{1,2} and is in line with item 22 of the PRISMA statement.³ References 1. Moons KGM, Wolff RF, Riley RD, et al. PROBAST: A Tool to Assess Risk of Bias and Applicability of Prediction Model Studies: Explanation and Elaboration. Ann Intern Med 2019;170(1):W1-w33. 2. Wolff RF, Moons KGM, Riley RD, et al. PROBAST: A Tool to Assess the Risk of Bias and Applicability of Prediction Model Studies. Ann Intern Med 2019;170(1):51-58. 3. Moher D, Liberati A, Tetzlaff J, Altman DG; PRISMA Group. Preferred Reporting Items for Systematic reviews and Meta-Analyses: the PRISMA statement. Ann Intern Med. 2009;151:264-9. [PMID: 19622511]
Figure 3: these are too low of quality to read – can you please send these in higher resolution, and then I will make my final comments on this section.	Please accept our apologies. Figure 3 has been revised and uploaded (for higher resolution) in PDF Format (as advised by the BMJ Editorial Production Assistant team).
Throughout: I'm used to seeing multivariable, not multi-variable	Thank you for your suggestion. We have amended the text throughout to read 'multivariable'.

General comments/edits:	
1. I would like to hear a bit more about the existing overlapping existing reviews mentioned in the introduction on line 41 – a short summary of what they found would be useful here.	Thank you for raising this point. We have added a short summary and amended the text as follows: ‘The current review extends and updates three broadly overlapping existing reviews.^{10 12 13} Whilst these reviews identified the use of various (derived and validated) RAMs for VTE in hospitalised patients, they did not find any evidence to suggest which RAM was superior. The aim of this systematic review was to identify primary validation studies (as derivation studies may give an overoptimistic assessment of model performance measures) and determine the accuracy of individual RAMs for predicting the risk of developing VTE in hospital inpatients.’
2. I agree with not performing a meta-analysis on this study cohort, primarily because the definition of the outcome and the included patient population differed so widely between studies. However, it would be of interest to see (descriptively) if certain characteristics were more common in studies with lower c-statistics, such as: higher author defined risk of bias, internal vs. external validation, broader age range of patients. (maybe others too, I am a statistical reviewer so don't know the full clinical implications of this, so open to other ideas). Or if you already have examined this, state in the methods/discussion.	Thank you for raising this point. Whilst we welcome your suggestion to explore and explain the varied C-statistics data (28 studies with 58 data points), we advise caution in presenting such data as the use of thromboprophylaxis (ranged from 3.8% to 100% in 25 studies) may lead to underestimation of predictive accuracy of a model if a given RAM were to predict VTE events that were subsequently prevented by thromboprophylaxis. In addition, limited reporting of thromboprophylaxis use (23 studies did not report on thromboprophylaxis use) may preclude further analysis of its impact upon the performance of the RAMs (a point which has already been noted in the discussion). On the other hand, if further exploration and presentation of the data is required by the reviewer, (our post hoc analysis did not reveal any noticeable differences within c-statistic groups and studied populations, study design, VTE definition, author defined risk of bias, validation method, age and sex) we are happy to report this.
3. Discussion: what are your specific recommendations do you have to design/implement a comprehensive study to determine the best RAM in this population? I appreciate your discussion on the limitations of	Thank you for your suggestion. We have amended the discussion section and have provided suggestions for future research

the studies presented here, but a bit more detail on how to carve a path forward would be beneficial.	
---	--

VERSION 2 – REVIEW

REVIEWER	Khalafallah, Alhossain Specialist Care Australia, Medicine and Clinical Haematology
REVIEW RETURNED	21-Mar-2021

GENERAL COMMENTS	The authors addressed adequately most of the issues that have been raised in the review process
---

REVIEWER	Campbell, Kristen University of Colorado, Pediatrics
REVIEW RETURNED	19-Mar-2021

GENERAL COMMENTS	Thank you for your thorough revision of this paper. I appreciate your comments, and only have some optional comments for improving Figure 3: - label each figure with "medical", "surgical" or "trauma", as appropriate - make all c-stats/confidence intervals that correspond to the same model the same color - the figures appear stretched and blurry, so fix these issues before publication
---

VERSION 2 – AUTHOR RESPONSE

Reviewer #1	
The authors addressed adequately most of the issues that have been raised in the review process	Thank you for your in-depth review and your comments on the work.

Reviewer #3	
Thank you for your thorough revision of this paper. I appreciate your comments, and only have some optional comments for improving Figure 3:	Thank you for your in-depth review, and your positive comments on the conduct, standard and writing of our work.

- label each figure with "medical", "surgical" or "trauma", as appropriate	Thank you for your comments. As suggested, we have amended the labelling for each figure in Figure 3.
- make all c-stats/confidence intervals that correspond to the same model the same color	Thank you for your comments. As suggested, we have amended the data (i.e. c-stats/confidence intervals) in Figure 3 so that the same models correspond to the same colour.
- the figures appear stretched and blurry, so fix these issues before publication	Thank you for your comments. As suggested, we have reconfigured Figure 3.